# Feeding Behavior, Gut Microbiota, and Transcriptome Analysis Reveal Individual Growth Differences in the Sea Urchin *Strongylocentrotus intermedius*

**DOI:** 10.3390/biology13090705

**Published:** 2024-09-07

**Authors:** Qi Ye, Chuang Gao, Haoran Xiao, Shuchao Ruan, Yongjie Wang, Xiaonan Li, Yaqing Chang, Chong Zhao, Heng Wang, Bing Han, Jun Ding

**Affiliations:** Key Laboratory of Mariculture and Stock Enhancement in North China’s Sea (Ministry of Agriculture and Rural Affairs), Dalian Ocean University, Dalian 116023, Chinahanbing@dlou.edu.cn (B.H.)

**Keywords:** growth differences, feeding behavior, gut microbiota, transcriptome

## Abstract

**Simple Summary:**

The presence of individual growth differentiation within the sea urchin (*Strongylocentrotus intermedius*) has significantly hindered the advancement of aquaculture. Despite this, the underlying causes and mechanisms driving these growth disparities remain largely unexplored. Here, we provide evidence that sea urchins of different sizes differ to varying degrees in behavior, gut microbiota compositions, and gene transcription profiles, which may collectively elucidate the observed growth disparities. Firstly, larger sea urchins demonstrated significantly heightened feeding activity compared to smaller sea urchins. Secondly, analysis of the gut microbiota revealed that larger sea urchins possess enhanced digestive efficiency and greater growth potential. Finally, the transcriptome research results prove that small-sized sea urchins were in a state of long-term stress and had some physiological responses to counteract the stress. In conclusion, we have elucidated the potential factors contributing to growth disparities in sea urchins across three distinct levels. This work provides a theoretical basis for regulating the growth of sea urchins, exploring methods to enhance aquaculture production, and promoting the sustainable and healthy development of the sea urchin aquaculture industry.

**Abstract:**

Growth differentiation among farmed sea urchins (*Strongylocentrotus intermedius*) poses a significant challenge to aquaculture, with there being a limited understanding of the underlying molecular mechanisms. In this study, sea urchins with varying growth rates, reared under identical conditions, were analyzed for feeding behavior, gut microbiota, and transcriptomes. Large-sized sea urchins demonstrated significantly higher feeding ability and longer duration than smaller ones. The dominant phyla across all size groups were Campylobacterota, Proteobacteria, and Firmicutes, with Campylobacterota showing the highest abundance in small-sized sea urchins (82.6%). However, the families Lachnospiraceae and Pseudomonadaceae were significantly less prevalent in small-sized sea urchins. Transcriptome analysis identified 214, 544, and 732 differentially expressed genes (DEGs) in the large vs. medium, large vs. small, and medium vs. small comparisons, respectively. Gene Ontology and KEGG pathway analyses associated DEGs with key processes such as steroid biosynthesis, protein processing within the endoplasmic reticulum, and nucleotide sugar metabolism. Variations in phagosomes and signaling pathways indicated that size differences are linked to disparities in energy expenditure and stress responses. These findings provide a foundation for future investigations into the regulatory mechanisms underlying growth differences in *S. intermedius* and provide clues for the screening of molecular markers useful to improve sea urchin production.

## 1. Introduction

The sea urchin *Strongylocentrotus intermedius* has the characteristics of short spines, rapid growth, and good gonad quality, and is well-known for its nutritional value, making it a popular species among the edible sea urchin species. Consequently, it is the most economically important cultured sea urchin in China. However, among echinoderms with economic importance, especially sea cucumbers and sea urchins [1,2], the phenomenon of growth differentiation is prominent and generally much higher than seen in other cultured invertebrates or fishes [3]. Individual growth differentiation among farmed sea urchins often leads to a series of problems, including decreased breeding efficiency, a prolonged breeding cycle, and differences in breeding specifications, which ultimately affect the benefits of the sea urchin industry. Therefore, analyzing the possible causes of excessive growth differences during farming and exploring techniques to achieve relatively balanced yet rapid sea urchin growth is of great significance to the industry.

The growth of organisms is a complex process, and growth differences inevitably exist between different populations of the same species and between different individuals within the same population. A study of sea cucumbers found that individuals with a faster growth rate exhibited a longer feeding time and higher feeding activity [4]. Additionally, large-bodied whiteleg shrimp *Litopenaeus vannamei* spent more time feeding than small-bodied individuals [5]. Similar results have been found in studies of rainbow trout *Oncorhynchus mykiss* [6]. The intestinal flora provide the host with a variety of physiologically active substances, and the microbiome plays an important role in promoting the body’s metabolism and absorption of nutrients [7]. In a study of American eels (*Anguilla rostrata*), significant differences were observed in the composition of intestinal flora between fast-growing and slow-growing groups [8]; in a study of discus fish (*Symphysodon haraldi*), while no significant differences in intestinal flora were detected between fast-growing and slow-growing groups, the higher richness of specific bacterial species may be a critical factor contributing to growth disparities [9]. Similarly, investigations of individual growth differences in various aquaculture species, such as Japanese sea cucumber *Apostichopus japonicus* [7], grass carp *Ctenopharyngodon idella* [10], and large yellow croaker *Larimichthys crocea* [11], have demonstrated that the composition of intestinal flora is, importantly, partly responsible for growth differences among individuals reared under the same conditions. Furthermore, transcriptome studies of Nile tilapia *Oreochromis niloticus* [12], Manila clam *Ruditapes philippinarum* [13], giant river prawn *Macrobrachium rosenbergii* [14], and Chinese red perch *Siniperca chuatsi* that exhibited different growth rates have shown that the energy budget and lipid metabolism also play roles in growth differences [15]. However, previous studies on individual growth differences in farmed sea urchins have mostly focused on environmental factors [16,17], rearing density [18], food sources [19], and feeding methods [20]. This study looked at farmed sea urchins *S. intermedius* as the research object. We explored feeding behavior, intestinal flora, and a genome-wide transcription map as aspects that might explain the growth rates and growth differentiation of individual sea urchins in three different size categories. The results provide new insights and a basis for further analysis of growth differences in farmed sea urchins.

## 2. Materials and Methods

### 2.1. Animal Breeding and Experimental Design

The samples were obtained from the Key Laboratory of Mariculture & Stock Enhancement in North China’s Sea, Ministry of Agriculture and Rural Affairs (121.56° E, 38.87° N). Six hundred sea urchins *S. intermedius* from the same lineage, each exhibiting test diameters of 22.03 ± 1.85 mm, test heights of 13.48 ± 1.14 mm, and body weights of 4.03 ± 0.57 g, with statistically insignificant differences in test diameter (*p* > 0.05), were selected for a 45-day breeding experiment. The sea urchins were cultured in a 400 L tank (Length: 1295 mm, Width: 860 mm, Height: 457 mm), maintained under controlled conditions: temperature at 20 ± 2 °C, pH of 8.04 ± 0.20, salinity at 30 ± 1PSU (Practical salinity units), and dissolved oxygen concentration at 8 ± 0.5 mg/L. Daily maintenance included the removal of sea urchin feces via the siphon method, accompanied by the replacement of more than half of the tank’s water to ensure optimal water quality. Fresh *Ulva lactuca* Linnaeus was administered daily to maintain nutritional standards, with surplus *U. lactuca* assessed the following day to ensure continuous adequate food supply. Following a 45-day breeding period, individuals were classified based on test diameter: those in the top 5% were categorized as the large size group (L, 39.72 ± 0.72 mm), those in the bottom 5% as the small size group (S, 22.68 ± 0.50 mm), and the remainder as the medium size group (M, 29.19 ± 0.38 mm), which was selected for further research. Statistical analyses were performed on both the test height and body weight of the sea urchins, as shown in Figure 1. Significant differences in test diameter, test height, and body weight were observed among the three sea urchin size groups, as evidenced by analysis of variance (ANOVA) (*p* < 0.05).

In the feeding behavior study, 30 sea urchins of varying sizes were selected and placed on one side of a 400 L square tank. The behavioral experiment setup was adapted from the study conducted by Chi et al. [21]. Sufficient quantities of *U. lactuca* were positioned on the opposite side of the tank. Observations and recordings were made at 20 min intervals to monitor the number of sea urchins adhering to *U. lactuca*. A total of three observations were carried out, with the experiment replicated three times to ensure reliability. Behavioral data from sea urchins of varying sizes were analyzed using a One-Way ANOVA. A *p*-value of less than 0.05 was considered to be statistically significant. The analysis was conducted using SPSS version 23.0.

Analysis of gut microbiota and transcriptome: Following the completion of the behavioral experiment, the sea urchins underwent a 24 h fasting period to ensure the purity of the intestinal samples and to minimize the impact of digestive matter on the experimental results. Three sea urchins from each size category—L, M, and S—were random selected for dissection along their peristomial membranes. Given that interdental muscles play a critical role in feeding behavior, influence energy acquisition, and exhibit sensitivity to environmental factors, we selected these muscles for transcriptome analysis. The interdental muscles were carefully scraped using a sterile scalpel and immediately preserved in liquid nitrogen. Subsequently, intestinal samples were meticulously collected, flash-frozen in liquid nitrogen, and stored at −80 °C for 16S RNA sequencing. The entire dissection procedure was conducted on a sterile clean bench.

### 2.2. DNA Extraction and High-Throughput Sequencing

Total genomic DNA was extracted from the samples utilizing the E.Z.N.A. Soil DNA Kit (Omega, Norcross, GA, USA), strictly adhering to the manufacturer’s protocols. The integrity of the extracted DNA was assessed via agarose gel electrophoresis, while the quantification of DNA concentration was performed using a NanoDrop spectrophotometer (Thermo Fisher Scientific, Waltham, MA, USA). High-throughput sequencing of the 16S rRNA V4 region was successfully executed using the Ion S5 XL platform (Novogene, Beijing, China).

### 2.3. Diversity Analysis and Functional Prediction

Amplicon sequencing data analysis was conducted utilizing QIIME2 (version 2019.7, https://qiime2.org, accessed on 18 September 2023). Single-end demultiplexed reads were processed to yield operational taxonomic units (OTUs) through denoising via the DADA2 plugin in QIIME 2 [22]. Taxonomic classification of sequences to OTUs was performed using a Naïve Bayes classifier, which was trained on the SILVA v132 99% OTU database (https://www.arb-silva.de, accessed on 25 September 2023) incorporating only the V4 region of 16S rRNA as reference sequences. OTUs affiliated with Mitochondria and Chloroplast, along with those representing less than 0.01% of the total, were systematically excluded from the dataset. A Venn diagram was constructed to delineate the shared and unique OTUs across samples.

A phylogenetic tree was constructed using FastTree to analyze phylogenetic diversity, as delineated in [23]. Prior to conducting diversity analyses, the feature table was rarefied to the minimum number of reads detected in the sample dataset. α- and β-diversity metrics, including Good’ s coverage, Chao1 index, Shannon diversity index, and Bray–Curtis distance, were computed utilizing the q2-diversity plugin. Principal coordinate analysis (PCoA) and analysis of similarity (ANOSIM) were employed to elucidate the dissimilarities in microflora structure, leveraging the Bray–Curtis distance.

### 2.4. RNA Preparation, and cDNA Library Construction

Transcriptomic sequencing was carried out by Shanghai Sangon Biotech Co., Ltd. (Shanghai, China). Three replicates were established for each concentration gradient, with one sea urchin being randomly selected from each replicate. RNA was extracted from sea urchin interdental muscles using the Trizol method (Ambion/Invitrogen, Austin, TX, USA). An RNA integrity assessment was conducted using the Agilent 2100 Bioanalyzer (Agilent Technologies, Santa Clara, CA, USA). Samples exhibiting an RNA Integrity Number (RIN) of 7 or higher were advanced to subsequent analyses. Library construction was performed using the TruSeq Stranded mRNA LTSample Prep Kit (Illumina, San Diego, CA, USA), adhering to the manufacturer’s instructions. Subsequently, these libraries were sequenced on the Illumina sequencing platform (HiSeqTM 2500 or Illumina HiSeq X Ten) (Illumina, San Diego, CA, USA), generating 125 bp/150 bp paired-end reads. All RNA clean data were submitted to the Short Read Archive (SRA) Sequence Database at the National Center for Biotechnology Information (NCBI) (Accession No. PRJNA1064022).

### 2.5. Quality Control and Gene Annotation

Raw data (raw reads) of fastq format were firstly processed through fastp software (version 0.19.7). In this step, clean data (clean reads) were obtained by removing reads containing adapter, reads containing ploy-N, and low-quality reads from raw data. Subsequently, an index of the reference genome was built using Hisat2 v2.0.5. Paired-end clean reads were aligned to the reference genome and annotated with HISAT2 version 2.0.5.

### 2.6. Differential Gene Expression Analysis

A difference comparison analysis between comparison combinations was performed using DESeq2 R package (1.20.0) (|log2(FoldChange)| ≥ 1 and *p* ≤ 0.05.), and Benjamini and Hochberg were used to correct and modify *p*-values. Due to the large number of genes, the false positive probability can be increased; therefore, padj is used to adjust the *p*-value of the set test in order to control the false-positive probability of all of the genes. DEGs were identified between library pairs (L versus M, L versus S, M versus S).

### 2.7. GO and KEGG Enrichment Analysis of Differentially Expressed Genes

A GO enrichment analysis of differentially expressed genes was implemented by the clusterProfiler R package, in which gene length bias was corrected. GO terms with corrected *p*-value less than 0.05 were considered significantly enriched by differential expressed genes. We used the clusterProfiler R package to test the statistical enrichment of differential expression genes in KEGG pathways. The objective of the KEGG pathway enrichment analysis was to identify novel functional genes and pathways and to elucidate the genes that influence growth differences in sea urchins.

### 2.8. RT-qPCR Validation

Six highly expressed DEGs were randomly selected for validation, utilizing the 18S gene as the reference. Reverse transcription of RNA from sea urchins of three different sizes was performed using the PrimeScript™ RT reagent Kit (TaKaRa, Kusatsu, Japan), following the provided protocol. Fluorescence quantitative analysis was conducted using the Light Cycler 96 system with the SYBR^®^ Premix Ex TAq™ kit (TaKaRa, Japan). The quantitative RT-PCR reaction mixture comprised a total volume of 20 µL, consisting of 2 µL of cDNA template, 10 µL of 2× SYBR Green Master mix (TaKaRa, Japan), 0.8 µL of each primer, and 6.4 µL of PCR-grade water. The thermal cycling conditions were set as follows: an initial denaturation at 95℃ for 30s, followed by 40 cycles. Annealing and elongation phases were conducted at 95℃ for 5s and 60 °C for 32s, respectively. Fluorescence quantification results were computed by employing the 2^−ΔΔCT^ algorithm. Each sample was independently assayed in triplicate.

## 3. Results

### 3.1. Sea Urchin Feeding Behavior at Different Sizes

In the feeding behavior experiment, the number of sea urchins of different sizes that attached to flat green macroalgae *U. lactuca*, commonly known as sea lettuce, which was added to the tank in sufficient or insufficient amounts, was recorded for 60 min. Regardless of whether sufficient or insufficient food was provided, the number of sea urchins in all three size groups that attached to the sea lettuce gradually increased as time progressed, although a significantly higher number of large-sized than small-sized sea urchins attached to the food source (*p* < 0.05). When food was sufficient, 30.18% of the large-sized sea urchins and only 6.60% of the small-sized sea urchins attached to the sea lettuce within the first 20 min, demonstrating a clear tendency of larger sea urchins to move quickly toward food. At 40 min, still only 8.24% of the small-sized sea urchins were attached, compared with 60.12% of the large-sized sea urchins, while the remaining large-sized sea urchins had moved nearer to the sea lettuce by this time. When food was insufficient, large-sized sea urchins still attached and fed, reaching a proportion of 72.28% at 60 min, whereas most of the small-sized sea urchins were observed adhering to the wall or on the bottom of the tank, and the proportion that attached to sea lettuce reached only 11.16% after 60 min (Figure 2b). In addition, regardless of whether there was sufficient food or not, the proportion of medium-sized sea urchins that attached to it within 60 min was between that of the other two size groups.

### 3.2. Composition of Intestinal Flora in the Different Size Groups

After 16S rRNA sequencing, the quality control and the Effective Tags report for the sequencing (Appendix A), as well as OTU dilution curves (Appendix A), showed that the reads of all samples tended to level off as the sequencing depth increased. A Venn diagram analysis showed only 181 OTUs shared by the large-sized and medium-sized sea urchins, which had 1284 and 1375 unique OTUs, respectively. The small-sized and medium-sized sea urchins shared only 250 OTUs, and their unique OTUs were 781 and 1360, respectively. There were only 145 OTUs shared by the small-sized and large-sized sea urchins, with, respectively, 781 and 823 OTUs unique to each (Figure 3).

Alpha-diversity metrics were calculated for each size group to evaluate the abundance and diversity of intestinal flora of sea urchins of different sizes. There were no significant differences between the values of the Chao1 index of species richness and the Shannon index of species diversity among the three size groups (*p* > 0.05), but there were differences in both indices among the groups. Values of the Chao1 index for the large-sized, medium-sized, and small-sized groups were 591.09, 563.43, and 383.62, respectively. The Shannon index of the large-sized group (5.15) was higher than that of the medium-sized (5.02) and small-sized (1.99) groups. Principal coordinate analysis (PCoA) showed that the intestinal bacterial communities of large-sized and small-sized sea urchins were significantly separated (Figure 4). The top ten phylum-level intestinal microbiota ranked by relative abundance in each size group of sea urchins are shown in Figure 5a. The dominant phyla in all three sizes of sea urchins were Campylobacterota, Proteobacteria, and Firmicutes, although their relative abundances were very different. The relative abundance of Campylobacterota (34.7%) and Firmicutes (25.0%) were both higher in large-sized sea urchins than in medium-sized sea urchins (29.4% and 19.5%, respectively). The relative abundance of Proteobacteria was higher in medium-sized sea urchins (39.5%) than in large-sized sea urchins (28.4%). Notably, in small-sized sea urchins, the relative abundance of Campylobacterota was as high as 82.6%, which is much higher than that of large-sized and medium-sized sea urchins, whereas the relative abundance of Firmicutes (3.3%) and Proteobacteria (9.5%) were very low. At the family level (Figure 5b), the dominant families in large-sized sea urchins were Arcobacteraceae (34.64%), Lachnospiraceae (15.43%), Pseudomonadaceae (7.90%), and Rhodobacteraceae (4.92%); in medium-sized sea urchins, they were Arcobacteraceae (29.40%), Vibrionaceae (10.69%), Rhodobacteraceae (9.65%), and Planococcaceae (4.22%); and in small-sized sea urchins, they were Arcobacteraceae (82.55%), Vibrionaceae (2.44%), Pseudomonadaceae (2.19%), and Lachnospiraceae (0.61%). The family Arcobacteraceae was dominant in all three size groups, but its abundance was significantly higher in small-sized sea urchins than in the other two size groups. However, the abundances of families Lachnospiraceae and Pseudomonadaceae were significantly less in small-sized sea urchins compared with large-sized sea urchins.

### 3.3. Quality of the Transcriptome Sequencing Data

In this study, three biological replicates were set up for each group of large-, medium-, and small-sized sea urchins. Each sample obtained more than 6.00G of data. After quality control, a total of 383,668,490 filtered sequences (98.22%) were obtained, approximately 58.39–65.81% of the raw reads remained as clean reads, and 54.90–60.23% of the clean reads were mapped to the reference genome sequence (*Strongylocentrotus intermedius*, JBGVUB000000000, not yet published). Base quality and composition analysis showed that the GC content of the samples ranged from 36.66% to 40.19%, with an error rate of ≤0.03, Q20 of ≥96%, and Q30 of ≥89% (Appendix A). These results indicate that the sequencing results could be used for subsequent analysis.

### 3.4. Annotation and Functional Characterization of Transcriptomes of Sea Urchins of Different Sizes

Differential gene screening showed that a total of 214 DEGs were screened out between group L (large-sized sea urchins) and group M (medium-sized sea urchins) (|log2(FC)|≥1 and *p* ≤ 0.05), of which 105 were upregulated and 109 were downregulated; 544 DEGs were screened out between group L and group S (small-sized sea urchins), of which 281 were upregulated and 263 were downregulated; and 732 DEGs were screened out between group M and group S, of which 373 were upregulated and 359 were downregulated (Figure 6).

To better understand the interaction between biological functions and DEGs, all DEGs (L vs. M, L vs. S, M vs. S) were annotated in GO terms. The GO terms were divided into three categories: Cellular component (CC), Biological process (BP), and Molecular function (MF). In this study, GO terms that were significantly enriched (*p* < 0.05) in the three categories were screened. For L vs. M, the top-five GO terms with the most abundant BP were proteolysis, microtubule cytoskeleton organization, cytoskeleton organization, lipid biosynthetic process, and microtubule-based process. The top four most abundant GO terms in CC were non-membrane-bounded organelle, intracellular non-membrane-bounded organelle, cytoplasm, and cytoplasmic part. The top four most abundant GO terms in MF were scavenger receptor activity, cargo receptor activity, peptidase activity, acting on L-amino acid peptides, and peptidase activity (Figure 7a). The most abundant GO terms in the three different categories (CC, BP, and MF, respectively) for L vs. S were as follows: proteolysis; extracellular region, extracellular region part, and extracellular space; and zinc ion-binding, cargo receptor activity and scavenger receptor activity (Figure 7b). For M vs. S, the most abundant GO terms in the three different categories, respectively, were as follows: proteolysis, Golgi vesicle-mediated transport; protein-containing complex, extracellular region, catalytic complex, extracellular region, endomembrane system; and peptidase activity, acting on L-amino acid peptides, peptidase activity, endopeptidase activity, serine-type peptidase activity, serine hydrolase, and serine-type endopeptidase activity (Figure 7c).

Next, we performed KEGG pathway analysis to determine the functions of DEGs and biological pathways involved in regulating growth differences, and thereby determined the top 20 significantly enriched KEGG pathways (Figure 8). The pathway ‘Protein processing in endoplasmic reticulum’ was significantly enriched in all three size groups of sea urchins (*p* < 0.05). At the same time, the pathway ‘Steroid biosynthesis’ was significantly enriched in both the L vs. M and L vs. S groups (*p* < 0.05). In addition, each control group also had its own unique significantly enriched pathway. In the L vs. M group, two unique pathways (‘Spliceosome’ and ‘Amino sugar and nucleotide sugar metabolism’) were significantly enriched. In the L vs. S group, only one pathway (‘Proteasome’) was significantly enriched. In the M vs. S group, four pathways (‘Phagosome’, ‘mRNA surveillance pathway’, ‘Pentose phosphate’, and ‘Biosynthesis of amino acids’) were significantly enriched.

### 3.5. qRT-PCR Verification

To confirm the accuracy of the DEGs identified in the RNA-seq expression analysis, we randomly selected seven genes from the DEGs screened in the mRNA library for verification. These genes may be related to growth differences, and primers were designed based on the sequences (Table 1). As shown in Figure 9, the RT-qPCR results were significantly correlated with the RNA-seq results (*p* < 0.05). Overall, the RT-qPCR results confirmed the RNA-seq results, indicating the reliability and accuracy of the expression analysis.

## 4. Discussion

### 4.1. Growth Differences Caused by Different Sea Urchin Feeding Behaviors

In a high-density closed-system artificial breeding environments, farmed aquatic animals will intensify their intraspecific competition owing to factors such as poor habitat, size disparities between individuals, insufficient feed, and population crowding [24,25,26]. In this study, we found that small-sized sea urchins mostly adhered to the bottom or wall of the tank, moved relatively slowly, and had a significantly lower rate of attachment to the food resource when compared with large-sized sea urchins. Similar results have been reported in studies of sea cucumbers [4,27]. Farmed animals obtain energy by ingesting the food provided, and therefore differences in their ability to obtain food because of their feeding behavior may be a reason for individual growth differences, as observed here in *S. intermedius*. Moreover, we observed that when food was insufficient, the number of feeding individuals of small-sized sea urchins was significantly lower than that of large-sized sea urchins. Smaller individuals are more likely to face starvation in this situation, and prolonged starvation will lessen the food conversion rate of the sea urchins, thereby exacerbating growth differences among the cultured animals [28]. Therefore, feeding behavior is probably a main factor responsible for individual growth differences in *S. intermedius*. Regardless of whether a sufficient amount of food was provided, larger sea urchins showed stronger feeding ability than smaller ones. A difference in access to food resources between large and small individuals is therefore reflected in the growth differences between individuals.

### 4.2. Differences in Intestinal Flora Composition Affect the Growth of Sea Urchins

The composition of intestinal flora has attracted much research attention because of its huge role in affecting the host’s metabolism, growth and development [29,30], digestion and absorption [31], and immune homeostasis [32]. One study found that *Sebastes schlegelii* cultured at low-density showed lower abundances of intestinal flora compared with fish cultured at high density [33]. The diversity of intestinal flora in diseased Atlantic salmon *Salmo salar* [34] and *Penaeus vannamei* [35] was also lower. In this study, intestinal flora composition differed significantly among sea urchins of different sizes.

Campylobacterota was a dominant phylum in the intestinal flora of all three sizes of sea urchins, a finding consistent with previous studies [36]. The proportion of Firmicutes in the intestinal tract was higher in large-sized sea urchins than in small-sized and medium-sized sea urchins. As the most common microorganism in the gut of aquatic animals, Firmicutes promotes efficient harvesting of dietary energy through carbohydrate fermentation [37,38]. Hence, differences in the sea urchin growth rates may be explained by a reduced abundance of Firmicutes in the microbiome, resulting in the impairment of nutrient absorption. In addition, the abundance of Proteobacteria was significantly higher in large-sized than in small-sized sea urchins. Similar results were also found in fast-growing sea cucumbers [4]. Notably, at the family level, Arcobacteraceae had a high abundance in the intestinal flora of all three size groups of sea urchins, but was significantly highest in small-sized sea urchins.

Previous studies have shown that the family Arcobacteraceae is widely present in the gut of an array of aquatic animals, such as Japanese puffer *Takifugu rubripes* [39] and Mediterranean mussel *Mytilus galloprovincialis* [40]. In our study, the abundance of this family in the intestinal flora was as high as 82.5% in small-sized sea urchins. In recent years, Arcobacteraceae has been identified as a novel pathogen associated with intestinal diseases in humans and animals [41]. Thus, a high abundance of Arcobacteraceae in the intestinal tract of smaller sea urchins may pose a greater risk of disease at this life stage and be detrimental to the sea urchins’ growth and development. Alternatively, the very high abundance of Arcobacteraceae in small-sized sea urchins might influence the diversity of their intestinal flora, causing imbalance in the homeostasis of intestinal flora, which in turn could slow their growth. The family Rhodobacteraceae was significantly more abundant in the large-sized and medium-sized sea urchins than in the small-sized sea urchins. Rhodobacteraceae has metabolic flexibility and can provide energy for the host [42]; in addition, Rhodobacteraceae can produce antibacterial active substances to thwart invasions of pathogenic bacteria, such as *Vibrio* [43,44]. The more-rapid growth rates of the medium-sized and large-sized sea urchins may be related to a higher abundance of Rhodobacteraceae in their gut. These bacteria increase energy acquisition and are beneficial to the individual’s health. The family Lachnospiraceae is present in the intestines of most healthy people and is involved in the metabolism of a variety of carbohydrates. This bacteria was significantly more abundant in the large-sized and medium-sized sea urchins than in the small-sized ones, and likewise may be a potentially beneficial bacterium to *S. intermedius* [45]. In summary, although the dominant phyla were similar in the intestinal flora of sea urchins of different sizes, their abundances were very different among the size groups. A high abundance of Campylobacterota in the gut of the large-sized sea urchins would accelerate their energy harvest. In contrast, a very high abundance of Arcobacteraceae in the small-sized sea urchins may make them more susceptible to diseases and might seriously affect the homeostasis of their intestinal flora. Clear compositional differences in the intestinal flora of sea urchins of different sizes suggest that intestinal flora may be an important factor leading to growth differences among individuals, even ones that were reared together.

### 4.3. Preliminary Study on Transcriptome Differences in Sea Urchins of Different Sizes

Stress is a protective mechanism to maintain collective homeostasis, but long-term excessive stimulation will have adverse effects on the body’s physiology. Previous studies found that *Oreochromis niloticus* under salinity stress and Darby’s sturgeon *Acipenser dabryanus* under starvation conditions showed significant changes in genes in the steroid synthesis pathway, which may be a stress response [46,47]. In our study, the expression of *cyp51* in the ‘Steroid biosynthesis’ pathway was significantly upregulated in medium-sized sea urchins compared with large-sized sea urchins. The same results also appeared in small-sized sea urchins. The *cyp51* gene specifically encodes lanosterol 14-α demethylase, which plays an important role in the steroid biosynthesis pathway. We speculate that the presence of larger sea urchins is a stress to smaller sea urchins, and the upregulation of *cyp51* is a response to this stress (such as that also which happens in response to unfavorable salinities or water pollutants). The significant upregulation of multiple genes in the ‘Pentose phosphate’ pathway (which includes the enzymes transaldolase/fructose-6-phosphate aldolase, NAD-binding domain of 6-phosphogluconate dehydrogenase, phosphoglucomutase/phosphomannomutase, and alpha/beta/alpha domain II) in the medium-sized sea urchins proves that the steroid biosynthesis process is more active in smaller individuals. Therefore, to resist the stress response caused by external pressure, smaller-sized sea urchins that have been under ‘stress’ for a long time may experience physiologically adverse effects which will hinder their growth. The stress behavior was reflected in clinging to the edge of the tank even in the presence of a food source.

Additionally, the sensitive expression of Hsp70 in animals under environmental stress is expected to be a biomarker of animal stress [48]. In our study, we found that Hsp70-related genes in medium-sized sea urchins were significantly upregulated in multiple pathways. This suggested that the medium-sized sea urchins were in a relatively long-term stress state, similar to the small-sized sea urchins, as reflected in the significant upregulation of DnaJ domain genes, indicating that the small-sized sea urchins were also in a certain state of stress. Thus, the cause of this stress state may be the presence of numerous large sea urchins, causing many of the smaller sea urchins to cluster together, resulting in crowding and stress [49,50]. Accordingly, the larger sea urchins in our feeding experiment had easier access to the food resource [51]. Such conditions can lead to the Matthew effect of accumulated advantages, and thus may result in growth differences. In addition, we found that hexokinase (HK) gene expression was significantly less in the large-sized sea urchins than in the medium-sized sea urchins. HK is the first key step in glycolysis and affects the speed and direction of the entire process of sugar metabolism [52]. In a study of sea cucumbers, it was found that the expression of HK in sea cucumbers was higher during the reproductive period [53]; this may indicate that, compared with large sea urchins, medium-sized sea urchins need to increase their carbohydrate metabolism to maintain continued growth and resource storage. The lower HK expression level of sea urchins may indicate that sea urchins have better energy homeostasis [54]. Another study found that *Oncorhynchus mykiss* liver proteasome activity was negatively correlated with growth rate [55]. In our study, multiple proteasome subunit genes of small-sized sea urchins were also significantly upregulated in the ‘Protein processing in endoplasmic reticulum’ pathway, which shows that the protein turnover activity is high in tissues of smaller sea urchins. This high turnover activity is a high-energy-consuming process, which implies a waste of energy. At the same time, the significant upregulation of multiple protein synthesis-related genes in the pathway ‘Protein processing in endoplasmic reticulum’ also proves that smaller sea urchins have high protein turnover activity (Calreticulin family, Plug domain of Sec61p), which may further explain the slow growth rate in the small-sized sea urchins. At the same time, the upregulation of proteasome subunit genes would suggest that small sea urchins are at risk of disease [56]. Interestingly, we found a significant enrichment of ‘Phagosome’ pathway genes in both the medium-sized and small-sized sea urchins. Phagocytosis, as the main defense method, plays an important role in the innate immunity of sea urchins [57], sea cucumbers [58], and other invertebrates. Previous studies have found that the ‘Phagocytosis’ pathway in diseased *S. intermedius* also exhibited significant enrichment [59]. The enrichment of the ‘Phagosome’ pathway may be a manifestation of sea urchins’ defense against pathogens to maintain cellular homeostasis; this may imply that smaller sea urchins may be more vulnerable to diseases, causing a greater risk of disease outbreaks among farmed animals.

The transcriptome results reveal significant differences between the small-sized and large-sized sea urchins, mainly in energy consumption and immune stress response. At smaller sizes, sea urchins have developed a stress response under the stress imposed by the presence of larger individuals. In our feeding behavior experiment, they appeared to avoid the large-sized sea urchins, reducing their own food intake, which would easily cause hunger, while they were challenged by the stress caused by overcrowding. Some of the stimulated physiological reactions increased the protein turnover activity in small-sized sea urchins, which would increase energy consumption. In addition, the significant expression of immune defense-related genes indicates that smaller sea urchins are at greater risk of disease.

## 5. Conclusions

This study found that large-sized sea urchins are generally more efficient in foraging than medium- and small-sized sea urchins. Regardless of whether the food resource is sufficient or insufficient, larger sea urchins can easily obtain more food. The low foraging efficiency of smaller sea urchins may be attributable to ‘stress’ caused by the presence of larger sea urchins. In response to this stress, small-sized sea urchins showed a weaker tendency to move toward the food, and many adhered to the wall of the tank. The transcriptome results proved that the small-sized sea urchins were indeed in a long-term state of stress, yet possessed some physiological responses to resist stress. These reactions undoubtedly increase the energy burden of smaller sea urchins, affecting their growth rates at different sizes and generally increasing the size differences among the sea urchins. In addition, an analysis of the intestinal flora of sea urchins of different sizes showed a significantly higher abundance of Rhodobacteraceae in large-sized compared to small-sized sea urchins, and the abundance of Arcobacteraceae was significantly highest in small-sized sea urchins. This may show that larger sea urchins benefit from an intestinal flora composition that benefits their digestion ability and growth potential. In contrast, the intestinal flora composition of smaller sea urchins may put them at greater risk of disease. In short, our investigation helps to explain several reasons for differences in the growth of *S. intermedius* individuals; the findings provide a theoretical basis on how to possibly regulate differential growth performance in this popularly farmed echinoderm, explore ways to improve aquaculture production, and promote the green and healthy development of the sea urchin industry.

## Figures and Tables

**Figure 1 biology-13-00705-f001:**
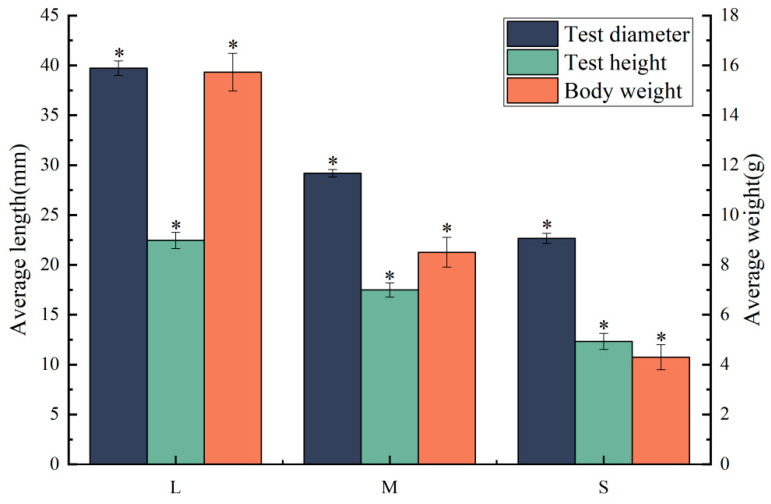
Different sizes of sea urchin test diameter, test height, and body weight. “*” indicates significant differences among different groups, *p* < 0.05.

**Figure 2 biology-13-00705-f002:**
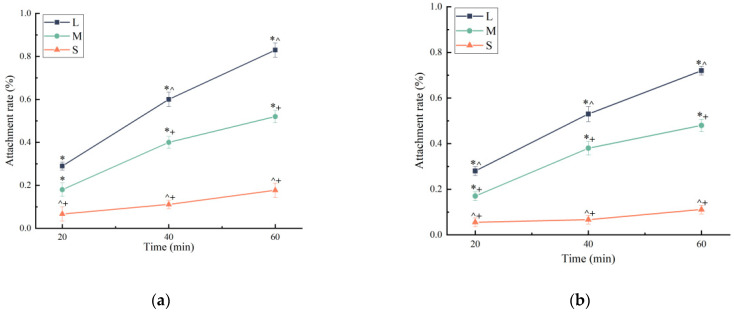
(**a**) Attachment rates of sea urchins of different sizes within 60 min when food is sufficient. (**b**) Attachment rates of sea urchins of different sizes within 60 min under food shortage conditions. (* *p* < 0.05 vs. S group; ^ *p* < 0.05 vs. M group; + *p* < 0.05 vs. L group).

**Figure 3 biology-13-00705-f003:**
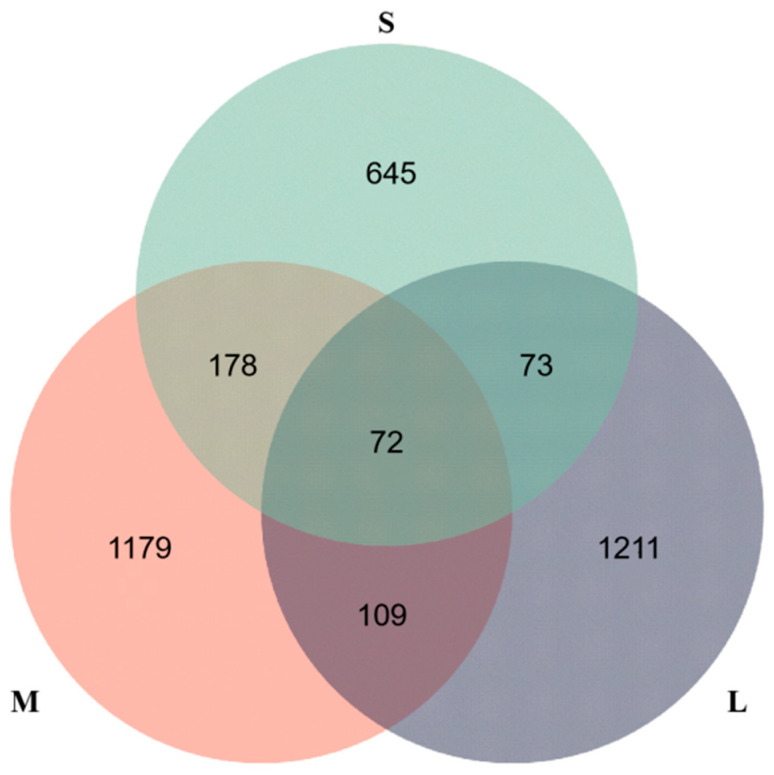
OTU number of gut microbiota communities of *S. intermedius* with different sizes.

**Figure 4 biology-13-00705-f004:**
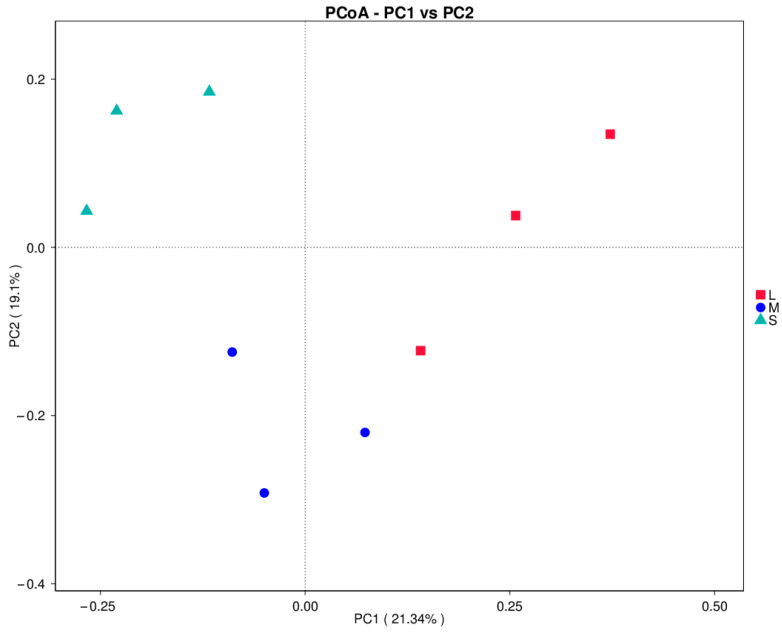
PCoA flora analysis.

**Figure 5 biology-13-00705-f005:**
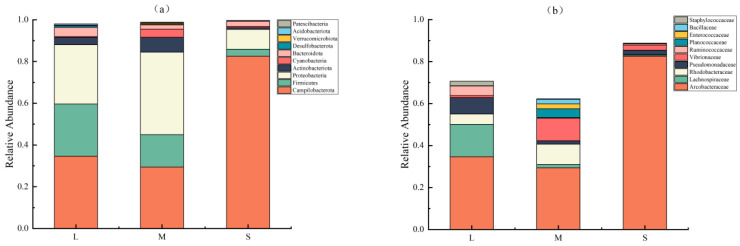
(**a**) Composition of gut microbiota at the phylum level of sea urchins of different sizes. (**b**) Composition of gut microbiota at the family level of sea urchins of different sizes.

**Figure 6 biology-13-00705-f006:**
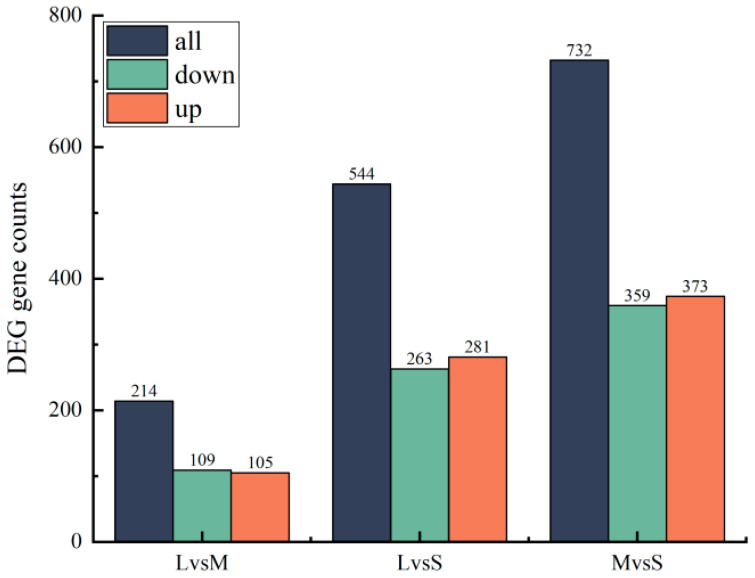
Histogram of DEGs in sea urchins of different sizes.

**Figure 7 biology-13-00705-f007:**
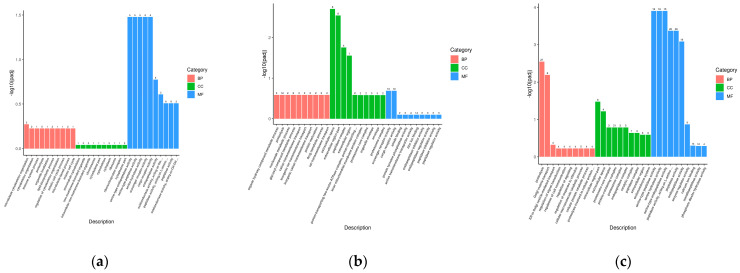
(**a**) GO enrichment analysis of the DEGs in L vs. M group; (**b**) GO enrichment analysis of the DEGs in L vs. S group; (**c**) GO enrichment analysis of the DEGs in M vs. S group.

**Figure 8 biology-13-00705-f008:**
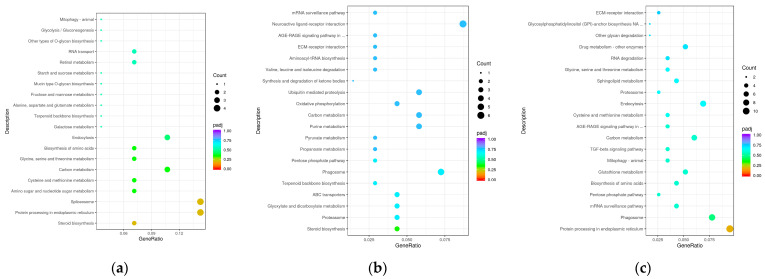
(**a**) KEGG pathway enrichment analysis of the DEGs in L vs. M group; (**b**) KEGG pathway enrichment analysis of the DEGs in L vs. S group; (**c**) KEGG pathway enrichment analysis of the DEGs in M vs. S group.

**Figure 9 biology-13-00705-f009:**
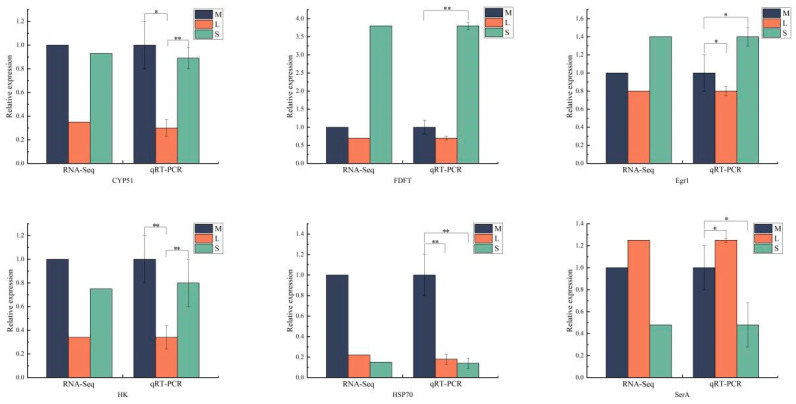
Verification of differential genes using qRT-PCR in different specifications of *S. intermedius.* “*” indicates significant differences among different group, *p* < 0.05, “**” indicates significant differences among different group, *p* < 0.01.

**Table 1 biology-13-00705-t001:** Primer sequences used in qRT-PCR.

Gene Name	Gene ID	Annealing Temperature (°C)	Sequence (5′ to 3′)
cyp51-F	XP_030851345.1	58	GAGAGATGACATTGGCAGCA
cyp51-R	XP_030851345.1	58	AGTGTGGAGGGAGTTTGACG
HK-F	XP_786955.2	60	ACTCCATCGTCTCCGAATGC
HK-R	XP_786955.2	60	CAACGCCTGCTACATGGAAG
Hsp70-F	XP_030845497.1	56	ACACTCATCTCGGAGGAG
Hsp70-R	XP_030845497.1	56	CTTTCTTATGCTTTCGCTTGA
Egr1-F	XP_030852777.1	59	TCCTGCGAGACGGCTTGTTTC
Egr1-R	XP_030852777.1	59	AGGTTGCTGGATGCGTATAGGC
FDFT-F	XP_030839983.1	59	GCAGCAGTCATTCAGGCATTGG
FDFT-R	XP_030839983.1	59	CAGTGTCCAGGGCTCTCAGAAC
SerA-F	XP_030841752.1	57	AGCACCGTTGAAGCACAGGAG
SerA-R	XP_030841752.1	57	GCGTGAGGGCGTTGGACAG
18S-F	XP_786791.2	56	GTTCGAAGGCGATCAGATAC
18S-R	XP_786791.2	56	CTGTCAATCCTCACTGTGTC

## Data Availability

The original contributions presented in the study are included in the article, further inquiries can be directed to the corresponding authors.

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
