# Peer review of "Feeding Behavior, Gut Microbiota, and Transcriptome Analysis Reveal Individual Growth Differences in the Sea Urchin Strongylocentrotus intermedius"

_biology, 2024, doi:10.3390/biology13090705_

Round 1

Reviewer 1 Report

Comments and Suggestions for Authors

Line 78, it says “Six hundred healthy sea urchins”, can the authors explain what is a “healthy sea urchin”? and how did they understand this? Or maybe it is not necessary to include the word “healthy”?

Line 89, change “Shell” by “Test” in all the manuscript.

Line 108, it says “24-hour fasting period.three sea” it needs punctuation.

Line 131, review the inter-word spacing

Line 314. Fix the figure legend.

Line 329. Use the correct punctuation in the figure legend.

Line 443. “Thus, the cause of this stress state may be the presence of numerous large sea urchins, causing many of the smaller sea urchins to cluster together, resulting in crowding and stress” is there a study that the authors can take reference of?

Comments on the Quality of English Language

Line 89, change “Shell” by “Test” in all the manuscript.

Author Response

Dear Editors and Reviewers:

Thank you for your letter and for the reviewers' comments concerning our manuscript entitled "Feeding behavior, gut microbiota, and transcriptome analysis reveal individual growth differences in the sea urchin Strongylocentrotus intermedius" (Submission lD: biology-3135985). Those comments are all valuable and very helpful for revising and improving our paper as well as the important guiding significance to our researches, We have studied comments carefully and have made correction which we hope meet with approval. Furthermore, we would like to show the details as follows:

Reviewer 1

Comments 1: Line 78, it says “Six hundred healthy sea urchins”, can the authors explain what is a “healthy sea urchin”? and how did they understand this? Or maybe it is not necessary to include the word “healthy”?

Response 1 : Thank you for pointing this out. The term "healthy" here refers to sea urchins with an undamaged appearance and rich in vitality. We agree with your comment and have therefore removed the term "healthy" from the manuscript (page 3, line 103). The revised sentence is: Six hundred sea urchins S. intermedius from the same lineage, each exhibiting test diameters of 22.03±1.85 mm, test heights of 13.48±1.14 mm, and body weights of 4.03±0.57 g, with statistically insignificant differences in test diameter (P>0.05), were selected for a 45-day breeding experiment.

Comments 2: Line 89, change “Shell” by “Test” in all the manuscript.

Response 2 : Thank you for your reminder. We have reviewed the entire manuscript and replaced all instances of "shell" with "test" (page 3, line 115). The revised sentence is: Following a 45-day breeding period, individuals were classified based on test diameter.

Comments 3: Line 108, it says“24-hour fasting period.three sea” it needs punctuation.

Response 3 : The error has already been corrected in the manuscript (page 4, line 138). The revised sentence is: Three sea urchins from each size category—L, M, and S—were random selected for dissection along their peristomial membranes. 

Comments 4: Line 131, review the inter-word spacing.

Response 4 : The error has been corrected in the manuscript, and the entire document has been thoroughly reviewed. Thank you for your meticulous review.

Comments 5: Line 314. Fix the figure legend.

Response 5 : Thank you for your reminder. The figure legend has been corrected, and the legends for other figures in the manuscript have also been checked (page 13, line 379).

Comments 6: Line 329. Use the correct punctuation in the figure legend.

Response 6 : Thank you again for your reminder. The punctuation has been corrected, and the entire manuscript has been reviewed for punctuation accuracy (page 11, line 335-337). The revised  legend is: Figure. 8 (a) KEGG pathway enrichment analysis of the DEGs in L vs M group; (b) KEGG pathway enrichment analysis of the DEGs in L vs S group; (c) KEGG pathway enrichment analysis of the DEGs in M vs S group.

Comments 7: Line 443. “Thus, the cause of this stress state may be the presence of numerous large sea urchins, causing many of the smaller sea urchins to cluster together, resulting in crowding and stress” is there a study that the authors can take reference of?

Response 7 : Thank you for your thoughtful reminder. After reviewing relevant literature, we have added other researchers' findings as a reference in this section (page 16, line 500). The added reference is:

[50] Vadas, R.; Elner, R.; Garwood, P.; Babb, I. Experimental evaluation of aggregation behavior in the sea urchin Strongylocentrotus droebachiensis: a reinterpretation. Marine Biology 1986, 90, 433-448, doi: https://doi.org/10.1007/bf00428567. 

[51]. Sun, J.; Zhao, Z.; Zhao, C.; Yu, Y.; Ding, P.; Ding, J.; Yang, M.; Chi, X.; Hu, F.; Chang, Y. Interaction among sea urchins in response to food cues. Scientific Reports 2021, 11, 9985, doi: https://doi.org/10.1038/s41598-021-89471-2. 

Comments 8: Line 89, change “Shell” by “Test” in all the manuscript.

Response 8 : Thank you for your reminder. We have reviewed the entire manuscript and replaced all instances of "shell" with "test".

Reviewer 2 Report

Comments and Suggestions for Authors

The manuscript provide a valuable information of the growth differences of farmed sea urchins Strongylocentrotus intermedius. It is a topic of interest to the researchers in the related area.

The authors should provide the clearly experimental design data, such as, salinity (30% ?) and dissolved oxygen concentration.

Author Response

Dear Editors and Reviewers:

Thank you for your letter and for the reviewers' comments concerning our manuscript entitled "Feeding behavior, gut microbiota, and transcriptome analysis reveal individual growth differences in the sea urchin Strongylocentrotus intermedius" (Submission lD: biology-3135985). Those comments are all valuable and very helpful for revising and improving our paper as well as the important guiding significance to our researches, We have studied comments carefully and have made correction which we hope meet with approval. Furthermore, we would like to show the details as follows:

Reviewer 2

Comments 1: The authors should provide the clearly experimental design data, such as, salinity (30% ?) and dissolved oxygen concentration.

Response 1 : Thank you for pointing this out. We agree with your opinion and have revised this section after reviewing the data(page 3, line 108-110). The revised sentence is: The sea urchins were cultured in a 400- liter tank(Length: 1295mm, Width: 860mm, Height: 457mm), maintained under controlled conditions: temperature at 20 ± 2°C, pH of 8.04 ± 0.20, salinity at 30 ± 1PSU(Practical salinity units), and dissolved oxygen concentration at 8 ± 0.5mg/L.

Reviewer 3 Report

Comments and Suggestions for Authors

This paper represents an intriguing insight on how the behavioral patters of sea urchins may be influenced by their gut microbiota. The behavioral experiments are good, and the results on the differential OTU diversity of gut flora are stark. However, I have one main concern that I think must be addressed, as it can affect the integrity of the claims made:

Line 162: For transcriptomic analyses, I think that some further quality control of the datasets are be necessary. Quality control is mentioned on Line 273, but not described. For example, using TransDecoder on the longest transcripts to identify only the coding regions, and using CD-Hit to cluster and yield a non-redundant set of transcripts. Without these steps, it is hard to trust the downstream analyses, e.g. do the non-coding regions bias the annotation software? Were isoforms or splice variants filtered out?

https://github.com/TransDecoder/TransDecoder
https://sites.google.com/view/cd-hit

Some of the results are also surprisingly neat - e.g. the 5% largest and smallest urchins all fall within very even size divisions; and in table A2, an error rate of 0.03% for each sample is surprisingly consistent. This might be a genuine representation of the results found, but it is a surprising pattern nonetheless.

Otherwise, I have only a few minor concerns:

Line 108: Why was the 24-hour fasting period necessary? Also, "period.three" should probably be "period. Three"

Line 149: Space needed before "Subsequently"

Line 303: Is "extracellular region" supposed to be written twice?

Overall, if my concern on the transcript quality is adequately addressed, I think this could be a nice paper.

Comments on the Quality of English Language

Paper is overall written well.

Author Response

Dear Editors and Reviewers:

Thank you for your letter and for the reviewers' comments concerning our manuscript entitled "Feeding behavior, gut microbiota, and transcriptome analysis reveal individual growth differences in the sea urchin Strongylocentrotus intermedius" (Submission lD: biology-3135985). Those comments are all valuable and very helpful for revising and improving our paper as well as the important guiding significance to our researches, We have studied comments carefully and have made correction which we hope meet with approval. Furthermore, we would like to show the details as follows:

Reviewer 3

Comments 1: Line 162: For transcriptomic analyses, I think that some further quality control of the datasets are be necessary. Quality control is mentioned on Line 273, but not described. For example, using TransDecoder on the longest transcripts to identify only the coding regions, and using CD-Hit to cluster and yield a non-redundant set of transcripts. Without these steps, it is hard to trust the downstream analyses, e.g. do the non-coding regions bias the annotation software? Were isoforms or splice variants filtered out?

Response 1 : Thank you for your thorough review of our manuscript. We performed quality control on the transcriptome data using the fastp software (version 0.19.7), followed by gene annotation based on the reference genome (not yet published). To address the issues raised, we have rewritten this section. Your expert feedback has been invaluable in helping me identify areas for improvement in my paper. Once again, I sincerely appreciate the time and effort you have dedicated to my research(page 5, line 186-205; page 10, line 321-323). The revised sentence is:

2.5. Quality control and gene annotation

Raw data (raw reads) of fastq format were firstly processed through fastp software. In this step, clean data (clean reads) were obtained by removing reads containing adapter, reads containing ploy-N and low quality reads from raw data. Subsequently, index of the reference genome was built using Hisat2 v2.0.5 Paired-end clean reads were aligned to the reference genome and annotated with HISAT2 version 2.0.5.

2.6. Differential gene expression analysis

Difference comparison analysis between comparison combinations was performed using DESeq2 R package (1.20.0) (|log2(FoldChange)| ≥1 and P≤0.05.), and Benjamini and Hochberg were used to correct and modify P values. Due to the large number of genes, the false positive probability can be increased; therefore, padj is used to adjust the p-value of the set test in order to control the false positive probability of all genes.  DEGs were identified between library pairs (L versus M, L versus S, M versus S).

After quality control, a total of 383,668,490 filtered sequences (98.22%) were obtained, approximately 58.39 %–65.81 % of the raw reads remained as clean reads, and 54.90 %–60.23 % of the clean reads were mapped to the reference genome sequence (not yet published). 

Comments 2: Some of the results are also surprisingly neat - e.g. the 5% largest and smallest urchins all fall within very even size divisions; and in table A2, an error rate of 0.03% for each sample is surprisingly consistent. This might be a genuine representation of the results found, but it is a surprising pattern nonetheless.

Response 2 : Thank you for your reminder. We have rechecked the data to ensure its accuracy.

Comments 3: Line 108: Why was the 24-hour fasting period necessary? Also, "period.three" should probably be "period. Three"

Response 3 : I apologize for the confusion. The reason for fasting the sea urchins for 24 hours was to empty the intestinal contents, ensuring the purity of the intestinal samples and minimizing the impact of digestive materials on the experimental results. This explanation has been added to the manuscript(page 4, line 136-138). The revised sentence is: Analysis of gut microbiota and transcriptome: Following the completion of the behavioral experiment, the sea urchins underwent a 24-hour fasting period to ensure the purity of the intestinal samples and to minimize the impact of digestive matter on the experimental results.

Comments 4: Line 149: Space needed before "Subsequently"

Response 4 : A space has been added in this section, and the entire manuscript has been rechecked. The revised sentence is: Subsequently, these libraries were sequenced on the Illumina sequencing platform (HiSeqTM 2500 or Illumina HiSeq X Ten), generating 125 bp/150 bp paired-end reads.

Comments 5: Line 303: Is "extracellular region" supposed to be written twice?

Response 5 : Apologies, this was a writing error. It should be "extracellular region part," and it has been corrected in the manuscript. Thank you for your reminder(page 12, line 354). The revised sentence is: The most-abundant GO terms in the three different categories (CC,  BP and MF, respectively) for L vs S were: proteolysis; extracellular region, extracellular region part, extracellular space; and zinc ion-binding, cargo receptor activity, and scavenger receptor activity

Reviewer 4 Report

Comments and Suggestions for Authors

Comments and Suggestions for Authors:

The manuscript titled "Feeding behavior, gut microbiota, and transcriptome analysis reveal individual growth differences in the sea urchin Strongylocentrotus intermedius" by Ye et al. is an original and well-executed study that explores the biological mechanisms driving growth variations in Strongylocentrotus intermedius. The experimental design is thoughtfully structured and carefully implemented. The statistical methods are appropriately selected and described with sufficient detail, aligning well with the data presented. The manuscript provides a comprehensive analysis of multiple factors, including feeding behavior, gut microbiota, and transcriptomic differences, which are thoroughly examined and discussed. The discussion is objective and offers plausible explanations for the results. Overall, this manuscript is of high quality and is worthy of publication in the journal "Biology" following minor revisions.

Specific Comments:

Introduction

Line 48: Consider whether "affects" or "affect" is more appropriate in context.

Lines 58-64: Including specific examples that link intestinal flora composition to biological growth differences could enhance clarity and reader understanding.

Materials and Methods

Line 78: Provide detailed information about the source of the experimental sea urchins.

Lines 101-105: The methods for studying sea urchins' feeding behavior are well-established. Please add references or expand on the protocols used in this experiment.

Lines 110-111: Explain the rationale behind selecting interdental muscle for transcriptome analysis.

Results

Line 199: Clarify which significance analysis method was employed, and indicate the significance in Figure 2.

Discussion

Lines 414-415: The discussion is insightful, but consider softening the conclusion with terms like "may be" to reflect the potential variability in the factors leading to growth differences.

References

Line 514: Ensure that Latin names of species (e.g., Apostichopus japonicus) are italicized.

Format

Ensure there is a space between the number and the unit.

Author Response

Dear Editors and Reviewers:

Thank you for your letter and for the reviewers' comments concerning our manuscript entitled "Feeding behavior, gut microbiota, and transcriptome analysis reveal individual growth differences in the sea urchin Strongylocentrotus intermedius" (Submission lD: biology-3135985). Those comments are all valuable and very helpful for revising and improving our paper as well as the important guiding significance to our researches, We have studied comments carefully and have made correction which we hope meet with approval. Furthermore, we would like to show the details as follows:

Reviewer 4

Comments 1: Line 48: Consider whether "affects" or "affect" is more appropriate in context.

Response 1 : Thank you for your reminder. We have replaced "affects" with "affect." (page 2, line 65) The revised sentence is: Individual growth differentiation among farmed sea urchins often leads to a series of problems, including decreased breeding efficiency, a prolonged breeding cycle, and differences in breeding specifications, which ultimately affect the benefits of the sea urchin industry.

Comments 2: Lines 58-64: Including specific examples that link intestinal flora composition to biological growth differences could enhance clarity and reader understanding.

Response 2 : We appreciate and accept your suggestion. We have added examples in the introduction section to enhance clarity and improve reader comprehension(page 2, line 77-82). The revised sentence is: In a study of American eels(Anguilla rostrata), significant differences were observed in the composition of intestinal flora between fast-growing and slow-growing groups [8]; in a study of discus fish(Symphysodon haraldi), while no significant differences in intestinal flora were detected between fast-growing and slow-growing groups, the higher richness of specific bacterial species may be a critical factor contributing to growth disparities [9]. The added reference is:

[8] Zhang, Y.T.; Huang, J.; Jin, W.; Ma, X.; Lv, G.; Ye, C.; Mu, J.; Wen, H.; Chen, S.X. Comparative analysis of gut microbiota and intestinal transcriptomic profile between fast-and slow-growing American eels (Anguilla rostrata). Aquaculture Reports 2024, 36, 102087, doi:https://doi.org/10.1016/j.aqrep.2024.102087.

[9] Zhang, Y.; Wen, B.; David, M.A.; Gao, J.-Z.; Chen, Z.-Z. Comparative analysis of intestinal microbiota of discus fish (Symphysodon haraldi) with different growth rates. Aquaculture 2021, 540, 736740, doi: https://doi.org/10.1016/j.aquaculture.2021.736740. 

Comments 3: Line 78: Provide detailed information about the source of the experimental sea urchins.

Response 3 : We have accepted your suggestion and added detailed information about the sea urchin sources in the Materials and Methods section(page 3, line 101-103). The revised sentence is: The samples were obtained from the Key Laboratory of Mariculture & Stock Enhancement in North China's Sea, Ministry of Agriculture and Rural Affairs (121.56° E, 38.87° N).

Comments 4: Lines 101-105: The methods for studying sea urchins' feeding behavior are well-established. Please add references or expand on the protocols used in this experiment.

Response 4 : The sea urchin behavior experiments were conducted with reference to the study by Chi et al., with appropriate adjustments. This information has been added to the manuscript(page 3, line 127-128). The revised sentence is: The behavioral experiment setup was adapted from the study conducted by Chi et al [21]. The added reference is: [21] Chi, X.; Hu, F.; Qin, C.; Huang, X.; Sun, J.; Cui, Z.; Ding, J.; Yang, M.; Chang, Y.; Zhao, C. Conspecific alarm cues are a potential effective barrier to regulate foraging behavior of the sea urchin Mesocentrotus nudus. Marine Environmental Research 2021, 171, 105476, doi: https://doi.org/10.1016/j.marenvres.2021.105476.

Comments 5: Lines 110-111: Explain the rationale behind selecting interdental muscle for transcriptome analysis.

Response 5 : Given that interdental muscles play a critical role in feeding behavior, influence energy acquisition, and exhibit sensitivity to environmental factors, we selected these muscles for transcriptome analysis, We have added this information to the Materials and Methods section for clarification(page 4, line 139-142). The revised sentence is: Given that interdental muscles play a critical role in feeding behavior, influence energy acquisition, and exhibit sensitivity to environmental factors, we selected these muscles for transcriptome analysis.

Comments 6: Line 199: Clarify which significance analysis method was employed, and indicate the significance in Figure 2.

Response 6 : Thank you for your suggestion. We have added the significance analysis method used in this section to the Experimental Methods, and have also redrawn Figure 2 based on your recommendations(page 4, line 132-134; page 6, line 258). The revised sentence is: Behavioral data from sea urchins of varying sizes were analyzed using a One-Way ANOVA. A p-value of less than 0.05 was considered statistically significant. The analysis was conducted using SPSS version 23.0.

Comments 7: Lines 414-415: The discussion is insightful, but consider softening the conclusion with terms like "may be" to reflect the potential variability in the factors leading to growth differences.

Response 7 : We have accepted your suggestion and added "may be" in this part of the manuscript(page 15, line 467). The revised sentence is: Clear compositional differences in the intestinal flora of sea urchins of different sizes suggest that the intestinal flora may be an important factor leading to growth differences among individuals even though reared together.

Comments 8: Line 514: Ensure that Latin names of species (e.g., Apostichopus japonicus) are italicized.

Response 8 : Thank you for your thorough review. We have rechecked all Latin names throughout the manuscript and formatted them in italics.

Comments 9: Ensure there is a space between the number and the unit.

Response 9 : The entire manuscript has been reviewed, and the formatting has been corrected.